# Cost-Effective Bull’s Eye Aperture-Style Multi-Band Metamaterial Absorber at Sub-THz Band: Design, Numerical Analysis, and Physical Interpretation

**DOI:** 10.3390/s22082892

**Published:** 2022-04-09

**Authors:** Zohreh Vafapour

**Affiliations:** 1Department of Electrical and Computer Engineering, Queen’s University, Kingston, ON K7L 3N6, Canada; z.vafapour@queensu.ca or z.vafa@ucmerced.edu or z.vafapour@gmail.com; 2Department of Physics, School of Natural Sciences, University of California Merced, Merced, CA 95343, USA

**Keywords:** sensors, metasurface, high-power THz sources, semiconductor device, THz absorber, cost-effective optical absorber, ultra-narrow N-beam absorber, multi-band structures

## Abstract

Theoretical and numerical studies were conducted on plasmonic interactions at a polarization-independent semiconductor–dielectric–semiconductor (SDS) sandwiched layer design and a brief review of the basic theory model was presented. The potential of bull’s eye aperture (BEA) structures as device elements has been well recognized in multi-band structures. In addition, the sub-terahertz (THz) band (below 1 THz frequency regime) is utilized in communications and sensing applications, which are in high demand in modern technology. Therefore, we produced theoretical and numerical studies for a THz-absorbing-metasurface BEA-style design, with N-beam absorption peaks at a sub-THz band, using economical and commercially accessible materials, which have a low cost and an easy fabrication process. Furthermore, we applied the Drude model for the dielectric function of semiconductors due to its ability to describe both free-electron and bound systems simultaneously. Associated with metasurface research and applications, it is essential to facilitate metasurface designs to be of the utmost flexible properties with low cost. Through the aid of electromagnetic (EM) coupling using multiple semiconductor ring resonators (RRs), we could tune the number of absorption peaks between the 0.1 and 1.0 THz frequency regime. By increasing the number of semiconductor rings without altering all other parameters, we found a translation trend of the absorption frequencies. In addition, we validated our spectral response results using EM field distributions and surface currents. Here, we mainly discuss the source of the N-band THz absorber and the underlying physics of the multi-beam absorber designed structures. The proposed microstructure has ultra-high potentials to utilize in high-power THz sources and optical biomedical sensing and detection applications based on opto-electronics technology based on having multi-band absorption responses.

## 1. Introduction

Optical electromagnetic (EM) absorbers are progressively demanding because of their valuable applications in solar energy harvesting [1], thermal emission tailoring [2], and biomedical detecting [3]. In recent years, the design, measurement, and fabrication of resonant optical EM absorbers (OEMA) have received much attention [1,2,3,4,5,6]. Most OEMAs provide only within a single frequency [7], or a fixed spectral range absorption [8], which greatly limits their practical applications in spectroscopic detection and phase imaging. Meanwhile, multi-band and tunable absorbers are in demand. In the past few years, researchers have demonstrated a variety of OEMA components, while most were in the form of supporting a single [4,7] or double [6,9] resonant wavelength-absorbing performance. All OEMAs have interesting applications in the terahertz (THz) bands [10,11,12,13], microwave [14,15,16], and visible [17,18,19] frequency regime. However, single-band absorbers do not sufficiently meet the needs of a variety of applications. Therefore, multi-band absorbers have also been designed in GHz [20], THz [21,22,23,24,25], near-infrared (NIR) [3,26,27], visible [28,29], far-infrared (FIR) [30], mid-infrared (MIR) [31,32], and infrared (IR) [33,34,35] ranges.

The THz band covers the distance between the IR and millimeter waves [36,37,38] and has many applications [39,40,41,42,43]. Therefore, THz EM waves have been less discovered than those in the continuous spectrum regime. Imaging with THz radiation is appealing for security [44,45,46] and biomedical applications [47,48,49,50,51] due to its ability to penetrate most dry and nonpolar materials without harming the body cells [49]. There have been lots of reports on the excitation of surface plasmon resonances and magnetic plasmon resonances in metamaterial and metasurface designs in optical sensing applications [52,53,54,55,56,57]. Therefore, functional devices in the THz band have also become one of the recent research hotspots [37,58,59,60,61].

Associated with metasurface research and applications, it is essential to facilitate metasurface designs to be of the utmost flexible properties. Most of the multi-band THz absorber designed structures work at frequencies over 1.0 THz [22,23,24,25,62], while most of the high-power THz sources, based on opto-electronics technology, operate at the frequency range of the sub-THz band; therefore, it is essential to design THz absorber structures that have optical absorbance peaks in the frequency range below 1 THz.

In this article, we produce theoretical and numerical results for the THz-absorbing-metasurface bull’s eye aperture (BEA)-style [63] design with single- and N-beam absorption peaks. The potential of BEA as the device elements has been well recognized. Some work has been performed to understand how such bull’s eye structures enhance and focus the optical properties of the system [64]. In this study, we focused on the excitation of electric and magnetic surface plasmons in the proposed bull’s eye-style device to achieve high absorption at different frequencies at the sub-THz band, which has potential applications in sensitive and selective optical biomedical sensing and detection, THz sources, etc. We proposed a THz absorber design constituting of a multi-layer metasurface nested with N-semiconductor thin rings working below 1 THz. Through the aid of EM coupling with multiple semiconductor rings, we wanted to reach the number of absorption peaks between 0.1 and 1.0 THz. By increasing the number of semiconductor rings without altering any other parameters, we will show the translation trend of the absorption frequencies. In addition, we validate our spectral response results using EM field distributions and surface currents. Furthermore, we mainly discuss the source of the N-band absorber, the underlying physics, and the effect of each parameter on the absorption peak. Therefore, our proposed microstructure operates as a multi-band optical absorber which has an N-band absorption peak resonance below a 1 THz frequency regime. This proposed device can be utilized as a high-power THz source.

## 2. Structure Design and Materials

The 2D and 3D graphic views of the proposed THz-absorbing-metasurface BEA-style unit cells are provided in Figure 1. The proposed BEA-style structure is designed on a 5 µm-thick layer of commercially available substrate ($t_{s}$) with a dielectric constant of $\epsilon_{glass}=2.4025$. The unit cell of the BEA-style structure has a compact size of 148 ($P_{x}$) × 148 ($P_{y}$) µm. The simulation is started with the first unit cell designed, called “*unit cell A*”, comprised of a semiconductor–dielectric–semiconductor (SDS) sandwiched layer. Unit cell A includes a conventional single ring resonator (RR) structure which includes one semiconductor RR antenna which works as a single-band absorber in the THz frequency regime. Then, we designed a two RR structure which operates as a dual-band THz absorber called “*unit cell B*”. Afterward, we designed three RRs which were called “*unit cell C*”, and so forth until the seven RRs, called “*unit cell G*”, were formed, which are displayed in Figure 1. All these BEA-style unit cells have the same side length of $P_{x}$ and $P_{y}$. The main gap/split width is considered as g = 5 µm for all gaps, and the semiconductor ring/strip width of w = 5 µm is used for all the simulations. The geometrical parameters of the BEA-style antenna as the inner/outer radius of RRs are presented in Table 1.

As illustrated in Figure 1, the designed SDS BEA-style microstructure consists of a layer of RR semiconductor antennas, which are placed in a bull’s eye aperture shape, and a mirror layer, separated by a magnesium fluoride ($MgF_{2}$) dielectric material as a buffer layer which works as a dielectric spacer. We used a semiconductor thick film with a thickness of ($t_{m}$) 20 µm and the ($t_{RR}$) 2 µm thickness of the semiconductor RRs (see Table 1).

Indium antimonide (InSb) is an economical/low-cost and commercially accessible material in the form of bulk padding and can be effortlessly found in extremely pure form. It is a semiconductor material with a direct and narrow band gap of $E_{g}=0.17\;eV$ at 300 K; it has a very small electron effective mass of $m_{e}^{*}\approx$ 0.014 $m_{e}$ [65] for electrons, where $m_{e}$ is the electron mass, and a small density of states in the conduction band. InSb is a low-power-consuming semiconductor as it works below 0.5 V. InSb has a variety of applications in optics and photonics, such as in forward-looking infrared (FLIR) imaging systems [66], temperature sensor devices [67], thermal image detectors [68], biomedical detection [49,69,70,71,72,73], thermo-optical applications [74], and thermal imaging cameras [75]. The highest carrier mobility, small effective mass of electrons, and small density of states among other semiconductor materials make it one of the best choices for thermo-optic and electric applications.

In the past few years, InSb has been the focus of various theoretical, numerical, and experimental research studies on the development of several types of tunable thermo-optic device applications, such as sensors [67,76], modulators [76], switches [77], and buffers [78].

However, knowledge of these features is based on the study of optical properties. The optical properties of InSb, to quantitatively explain the optical spectral responses, have been studied using a simple Drude model [65], which is presented in the following equations. The Drude model, applied to a semiconductor plasma, treats electrons and holes as free particles subject to random collisions at an energy-independent rate of 1/τ. The complex dielectric function can be represented by different models such as the Drude model [65] and Drude–Lorentz model [79]. Here, we use Drude model, which is commonly used for metals [80] and highly doped semiconductors [60,65]. The complex dielectric function ($\epsilon_{DM}$) is given by [65]: (1)$$\epsilon_{DM}\left(\omega \right)=\epsilon^{\prime }\left(\omega \right)+i\epsilon^{\prime \prime }\left(\omega \right)$$
where, $\epsilon_{DM}\left(\omega \right)$ is the relative permittivity, $\epsilon^{\prime }\left(\omega \right)$ is the real, and $\epsilon^{\prime \prime }\left(\omega \right)$ the imaginary, parts of the relative permittivity, respectively, which plotted in Figure 2 and can be calculated as:(2)$$\epsilon^{\prime }\left(\omega \right)=\epsilon_{\infty }-\frac{Ne^{2}\tau^{2}}{m_{e}^{*}\epsilon_{0}\left(1+\omega^{2}\tau^{2}\right)}$$
(3)$$\epsilon^{\prime \prime }\left(\omega \right)=\frac{Ne^{2}\tau }{m_{e}^{*}\epsilon_{0}\omega \left(1+\omega^{2}\tau^{2}\right)}$$
where $\epsilon_{\infty },\;N,\;e,\;\tau ,\epsilon_{0}$, and ω are the high frequency dielectric constant, the carrier density, electron charge, carrier relaxation time, free space permittivity, and angular frequency of the incident radiation, respectively. For InSb, $\epsilon_{\infty }$ is equal to 15.68 [65]. The carrier density *N* is calculated as follows: (4)$$N=5.76\times 10^{14}\;T\sqrt{T}\;e^{\left(-\frac{0.26}{2k_{B}T}\right)}$$

Here, *T* is the absolute temperature and $k_{B}$ is the Boltzmann constant. Equation (4) shows that the carrier density changes by changing the temperature; thus, the complex dielectric function, $\epsilon_{DM}\left(\omega \right),$ and, corresponding to that, the refractive index (RI) of InSb also change [notice to Equations (1)–(4)]. The carrier relaxation time, *τ*, is achieved by the relation:(5)$$\tau =\frac{m_{e}^{*}µ}{e}$$
where µ is the electron mobility. The carrier relaxation time is replaced with an easily measurable quantity which reduces the complexity of the dielectric function.

We, first, performed the numerical computations using the finite-difference time-domain (FDTD) method [81] to calculate the spectral responses using commercial software of the CST Microwave Studio; then, we analyzed the results by the investigation of the EM field distributions and surface currents.

## 3. Spectral Responses

Pursuant to the effective medium theory (EMT), metasurface can be described by the effective relative permittivity as:(6)$$\epsilon \left(\omega \right)=\epsilon_{Re}\left(\omega \right)+i\epsilon_{Im}\left(\omega \right)$$
and the effective relative permeability as:(7)$$µ\left(\omega \right)={µ}_{Re}\left(\omega \right)+i{µ}_{Im}\left(\omega \right)$$

According to the above-mentioned frequency-dependent optical parameters (i.e., ϵ(ω) and µ(ω)), the complex RI and effective impedance can be obtained through the following formulas:(8)$$n\left(\omega \right)=\sqrt{\epsilon \left(\omega \right)µ\left(\omega \right)\;}$$
(9)$$Z\left(\omega \right)=\sqrt{\frac{µ\left(\omega \right)\;}{\epsilon \left(\omega \right)\;}}$$
which are complex values and defined as:(10)$$n\left(\omega \right)=n_{Re}\left(\omega \right)+in_{Im}\left(\omega \right)$$
(11)$$Z\left(\omega \right)=Z_{Re}\left(\omega \right)+iZ_{Im}\left(\omega \right)$$

The absorption of the proposed microstructure at normal incidence is obtained by the following relation:(12)A(ω)=1−T(ω)−R(ω)

From the absorption equation (i.e., Equation (12), it is verified that the absorption is dependent on frequency-dependent parameters such as T(ω) and R(ω), which are transmission and reflection, respectively. They are calculated, as follows, using $S_{21}$ and $S_{11},$ which are the scattering parameters (S-parameters) relevant to transmission and reflection, respectively:(13)$$T\left(\omega \right)={\left|S_{21}\left(\omega \right)\right|}^{2}$$
(14)$$R\left(\omega \right)={\left|S_{11}\left(\omega \right)\right|}^{2}$$

The free-space transmission is calculated as: (15)$$S_{21}{\left(\omega \right)}^{-1}=\left[\sin\left(n\left(\omega \right)kd\right)-\frac{i}{2}\left(Z\left(\omega \right)+\frac{1}{Z\left(\omega \right)}\right)\cos\left(n\left(\omega \right)kd\right)\right]e^{ikd}$$
in which:(16)$$k=\frac{\omega }{c}$$
where *c* is the speed of light in the vacuum, *d* is the thickness of the slab, and Z(ω) is the effective impedance of the structure. As Z(ω) approaches one (i.e., free-space value, Z(ω) ≃ 1), the transmission entirely calculated by considering *n*(*ω*) is as follows:(17)$$S_{21}{\left(\omega \right)}^{-1}=\left[\sin\left(n\left(\omega \right)kd\right)-i\cos\left(n\left(\omega \right)kd\right)\right]e^{ikd}$$

Upon substitution of the exponential forms, Equation (17) becomes:(18)$$S_{21}{\left(\omega \right)}^{-1}=e^{i\left(n_{Re}\left(\omega \right)-1\right)kd}e^{n_{Im}\left(\omega \right)kd}$$

Based on the transmission formula in Equations (13) and (18), we achieve this relation for the transmission:(19)$$T\left(\omega \right)=\exp\left[-2n_{Im}\left(\omega \right)kd\right)]$$

Due to the fact that the proposed microstructure design is backed by a semiconductor thick film (i.e., mirror layer), the EM wave cannot transmit from the proposed microstructure [69], and as $n_{Im}\left(\omega \right)$ approaches infinity (for a given *d*),
(20)$$\underset{n_{Im}\to \infty }{\lim}T\left(\omega \right)=0$$
therefore,
(21)$$\left|S_{21}\left(\omega \right)\right|=0$$

The reflection can be calculated through its scattering parameter [82,83] as follows:(22)$$S_{11}\left(\omega \right)=\frac{Z\left(\omega \right)-Z_{0}}{Z\left(\omega \right)+Z_{0}}$$
where $Z_{0}$ represents the free-space impedance, with the value of one ($Z_{0}=1$) for the normalized impedance. Therefore, based on the reflection formula in Equations (14) and (20), we achieve the below relation for the reflection [82,83]:(23)$$R\left(\omega \right)={\left|\frac{Z\left(\omega \right)-1}{Z\left(\omega \right)+1}\right|}^{2}$$

The simulated spectral responses and the real and imaginary parts of the impedance as a function of frequency for *unit cell A* case are calculated and plotted (see Figure 3). As can be seen in Figure 3b, when the real part of the effective impedance is nearly one (i.e., $Z_{Re}\left(\omega \right)$ = 1.811) and the imaginary is exactly zero (i.e., $Z_{Im}\left(\omega \right)$ = 0) at a frequency of peak absorption resonance of $f_{1}$ = 1.6 THz, the reflection value decreases to zero (*R*(**ω** ≃ 0); so, the absorbance will close to one ($A_{max}$ = 0.9167), which means the effective impedances are nearly matched to the free space (see Figure 3).

This work started with the simulation of *unit cell A* that operates as a single-band THz absorber comprised of only one RR antenna SDS sandwiched layer, which shows one strong absorption peak in the THz frequency range at f = 1.6 THz (Figure 4 and Figure 5a). Moreover, we investigated the polarization-independent characteristic of the proposed design in Figure 4. As can be seen in Figure 4, there are no differences between the absorbance spectra in TE and TM polarization modes. Therefore, it can be concluded that our proposed design is a polarization-independent structure.

The concept of a metasurface THz absorber is particularly important at THz frequencies where it is difficult to find strong-frequency-selective THz absorbers. The single-band THz absorbers are valuable, but it is anticipated that they have some limitations in applications from the frequency selectivity aspect [4,7]. Therefore, to prevent these potential limitations, a multi-band design of the metasurface is urgently needed. However, in the fields of opto-electronics, THz communication systems [84], and photonics, it is necessary to obtain more information through multi-band and multi-beam means to develop multi-tasking systems, which produce potential applications for multi-band THz absorbers [85,86]. Therefore, we designed the second microstructure, defined as *unit cell B*, with two RR antennas which work as a dual-band THz absorber. As seen in row (b) of Figure 5, there are two absorption peaks in the blue-line curve (corresponding to the two dip resonances in the reflectance spectrum in the yellow-line curve) in the absorbance spectrum; one is perfect, with approximately 98.85% absorptivity at a frequency of 1.564 THz, and the second occurs at approximately 0.824 THz with approximately 74.64% absorptivity. In this way, the next proposed microstructures, called “*unit cell C/D/E/F/G*”, with three/four/five/six/seven RR antennas, respectively, were designed and simulated. As can be seen from row (g) of Figure 5, there are seven absorption peaks, and six of them are placed at the sub-THz band (under 1 THz frequency range) which is so valuable in opto-electronics technology and THz communication system applications. To have an N-band THz absorber with peak absorption in the sub-THz band regime, the designed microstructure should have N-ring resonators such as in the proposed design.

Most of the THz absorbers researched in the literatures operate at frequencies beyond 1.0 THz; however, since many high-power THz sources based on opto-electronics technology are working at a frequency range below 1.0 THz [22,23,24,25,62], it is essential to design a device that has optical absorption for both TE and TM polarization modes in the frequency range of the sub-THz band. Our multi-band proposed microstructure design operates below 1 THz, having N-band absorption peak resonances, which is so valuable in high-power THz sources and optical sensing and detection applications.

In the following, to reveal the physics behind the proposed design, we provide the electric and magnetic field energy density distributions of *unit cell G* with multi-beam absorption peak resonance frequency of the proposed THz absorber (Figure 6 and Figure 7) that has potential applications in spectroscopic detectors and imaging applications by utilizing high-power THz sources in the 0.1–1.0 THz frequency range.

## 4. Electromagnetic Field Distributions Discussion

Surface plasmon (SPs) are EM waves that travel along a metal/semiconductor/graphene/superconductor dielectric/air interface, in all frequency regimes based on the non-dielectric material that its interface has been exposed to, to the incident light. They are a type of surface wave, guided along the interface. SPs are shorter in wavelength than the incident light, which are known as photons, Therefore, they have subwavelength-scale limitations. The idiom, surface plasmon resonance (SPR), describes that the wave implies a charge motion in the non-dielectric material, which is known as surface plasmon. The EM wave resonances cause optical responses, which are known as resonance. SPR excites when polarized light incidences an electrically conducting surface at the interface between two different media, such as semiconductor and dielectric. This occurrence produces electron charge density waves, called plasmons, reducing the intensity of the reflected light.

To obtain an insight into the origin of the multi-beam absorption, we focused on the EM response of the case with the seven RR antennas (i.e., *unit cell G*) at normal incident light in TE/TM polarization modes (the proposed microstructure is a polarization-independent design, as can be seen in Figure 4). We defined every absorption peak resonance as an $f_{i}$ mode, where *i* refers to the specific RR antenna which is more excited in that resonance frequency; for example, the mode $f_{1}$ refers to the RR antenna with the radius of $r_{1}$ in Figure 1. We monitored the electric and magnetic fields and surface current densities on the top and bottom layers of the structure in the XY-plane at resonance frequencies of $f_{i}$ modes, as shown in Figure 6 and Figure 7, respectively.

As can be seen from Figure 6, we plotted the electric fields distributions and the excited surface plasmons as the surface current energy densities of the proposed microstructure in the XY-plane at different resonance frequencies. Column I in Figure 6 shows the electric field distribution at *Z* = 36 nm. Column II show the surface current electric energy densities, and column III shows the excited surface plasmons as the electric field distributions at *Z* = 25 nm. As clearly observable in row (a) of Figure 6, the innermost and smallest ring with a radius of $r_{1}$ is strongly coupled to the electric field at the $f_{1}$ mode of 1.55 THz. It supplies an independent electric dipole response (see Figure 6a, columns I to III), and the surface charge oscillates the most along the sides driven by the external electric field. Therefore, it can be concluded that in this case, $f_{1}$ mode and its corresponding absorption peak resonance is mostly due to the excitation of $r_{1}$ RR antenna. The magnetic component of the incident wave penetrates between the top and bottom layers and generates an antiparallel surface current on the RR antennas and the ground semiconductor plane, leading to the magnetic coupling and the response (see Figure 7a, columns I to III).

Figure 6 reveals that the absorption peak resonances originate from the dipole electric response of the RR antennas and the magnetic surface plasmon response between the air–semiconductor–dielectric (ASD) sandwiched layer. The first, second, third, to finally, seventh absorption peak resonance is associated with the EM resonance of the inner ring of $r_{1}$, $r_{2}$, $r_{3},$ to the outer ring of $r_{7}$, respectively. The resonant frequencies are inversely proportional to the radius of the RR antenna $r_{i}$ (*i* = 1 to 7), which indicates the bigger the radius of RR, the smaller the resonant frequency will be.

In the last part of this research, in order to show the effect of the geometrical parameter of the RR antennas on the absorption peak high and the frequency shift of the proposed microstructure, we provide the investigation of the *unit cell G* with the multi-beam absorption peak resonance frequency.

## 5. Studying the Absorption Peak Characteristics

As seen in Figure 5, in the case of *unit cell A*, there is only one resonance peak at around 1.6 THz; however, when we designed *unit cell B* with two RR antennas, there are two absorption peak resonances at approximately $f_{FAP}$ = 0.82 THz (frequency of first absorption peak) and $f_{LAP}$ = 1.56 THz (frequency of last absorption peak) with two different high absorptions. Therefore, we defined a parameter as the “working frequency duration (WFD)” for the frequency shift from the first to last peak absorption resonance for the proposed device:(24)$$\mathrm{WFD}=f_{FAP}-f_{LAP}$$

The WFD parameter for *unit cell A* should be zero because of the excitation of only one mode, due to the creation of one absorption peak resonance. As seen in Figure 8, the WFD value is the maximum for the case of *unit cell G* due to the excitation of the seven-beam absorption from a frequency of 0.2 THz to 1.55 THz. Therefore, it can be concluded that, by adding the number of RR antennas, the WFD increases due to excitation of higher order modes because of the existence of more RR antennas in the device.

We plotted Figure 9 to show the absorption percentage (corresponding to the absorption high) in any specific frequency for the case of *unit cell G* and investigated the strength of every RR antenna (with $r_{i}$ outer radius) corresponding to every mode ($f_{i}$). As clearly seen in Figure 9a,b, the first/last and highest/shortest absorption peak is approximately 97.8/10.5% which refers to the innermost/outermost RR antenna (i.e., the RR antenna with an outer radius of $r_{1}/r_{7}$) which occurs at approximately $f_{1/7}=1.55/0.2$ THz.

It can be concluded that, by increasing the number of RR antennas, the number of absorption peak resonance as well as the defined WFD parameter increase, and there is a higher absorption peak in the absorbance spectral response due to the excitation of the higher order modes which was created by having more semiconductor RR antennas in the system.

## 6. Conclusions and Future Directions

In this work, plasmonic interactions in device structures containing a polarization-independent semiconductor–dielectric–semiconductor (SDS) sandwiched layer design was analyzed. Based on this analysis, designed structures are defined which have wide potential applications in opto-electronics, terahertz (THz) communication, optical sensors, and detection in biomedical systems. This plasmonics-based approach leads to the conception of novel applications, including multi-band super-absorber micro-structures that are the basis for a variety of THz communication systems. We achieved a multi-beam THz absorber at a sub-THz band which has the potential to be utilized in high-power THz sources applications based on opto-electronics technology. The theoretical and numerical results were produced for a THz-absorbing metasurface bull’s eye aperture-style (BEA) design with N-beam absorption peaks at a sub-THz band frequency regime using economical and commercially accessible materials which have low-cost and easy fabrication processes for sensitive and selective sensing, detection, and other THz-related applications.

Future work involves the use of other types of semiconductors such as InAs, InP, GaAs, or with different and flexible optical properties. Different dielectric materials can also be used to tune the surface-plasmon frequency.

## Figures and Tables

**Figure 1 sensors-22-02892-f001:**
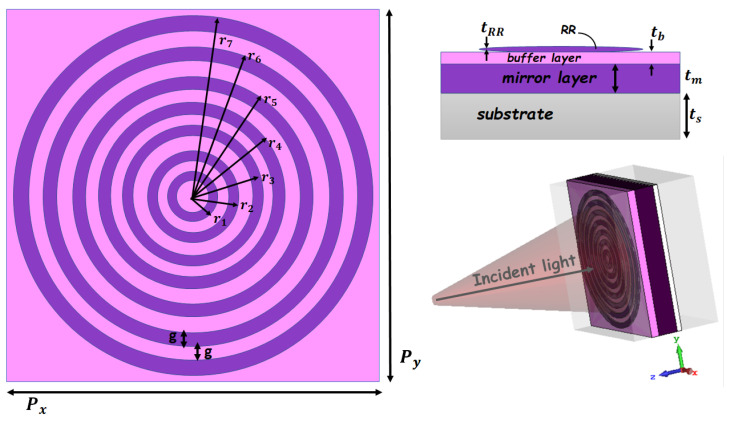
The 2D and 3D views of the proposed design (*unit cell G*).

**Figure 2 sensors-22-02892-f002:**
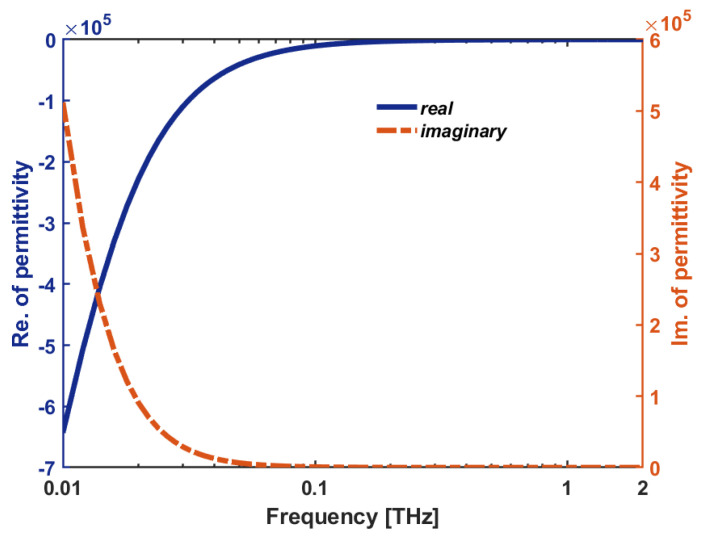
The complex dielectric function ($\epsilon_{DM}\left(\omega \right)$: electric permittivity) of InSb vs frequency.

**Figure 3 sensors-22-02892-f003:**
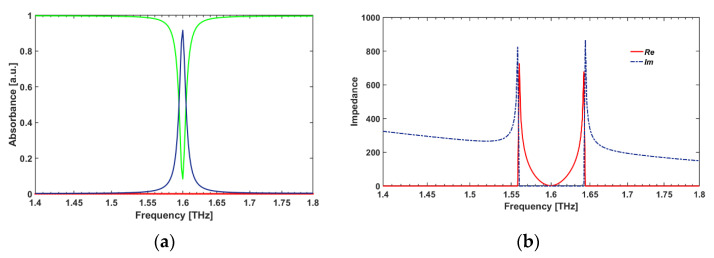
(**a**) The absorbance (blue-line), and the reflectance (green line) spectral responses, and (**b**) the complex effective impedance of *unit cell A* as a function of frequency.

**Figure 4 sensors-22-02892-f004:**
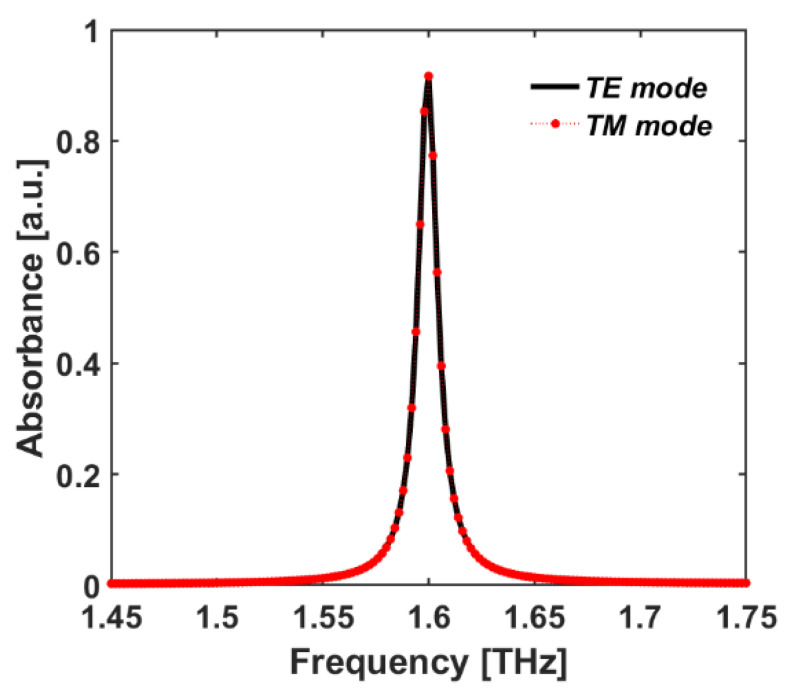
The absorbance spectrum of *unit cell A* for TE and TM polarization modes vs frequency.

**Figure 5 sensors-22-02892-f005:**
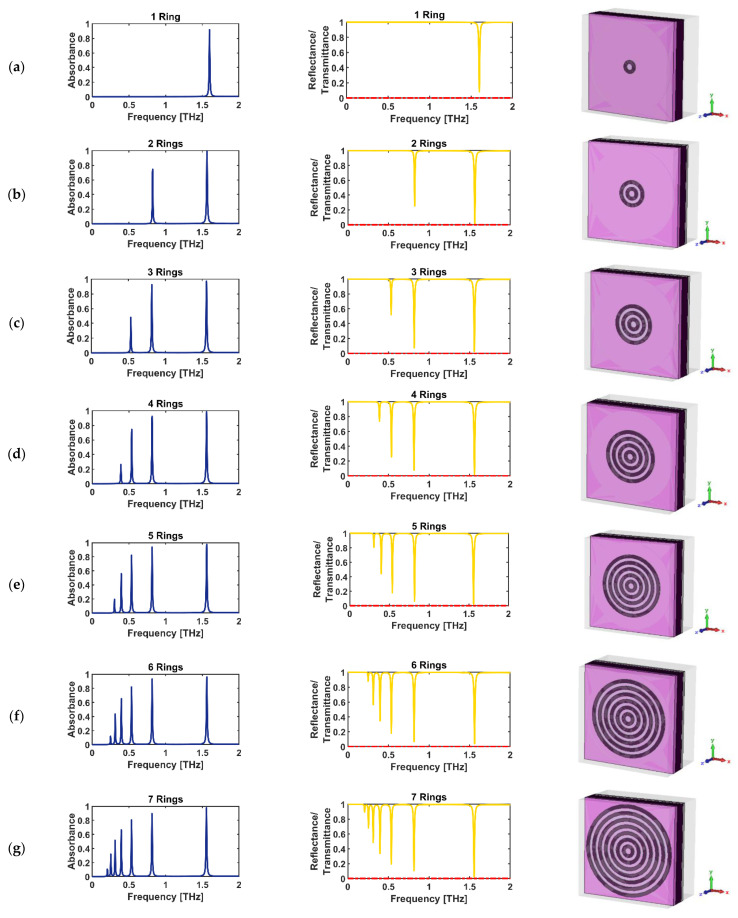
The left-side column shows the absorbance spectra (in blue); the middle column shows reflectance (in yellow) and transmittance (in red) spectra; and the right-side column shows the 3D illumination of the unit cell: (**a**) *A* with one RR antenna, (**b**) *B* with two, (**c**) *C* with three, (**d**) *D* with four, (**e**) *E* with five, (**f**) *F* with six, and (**g**) *G* with seven RR antennas.

**Figure 6 sensors-22-02892-f006:**
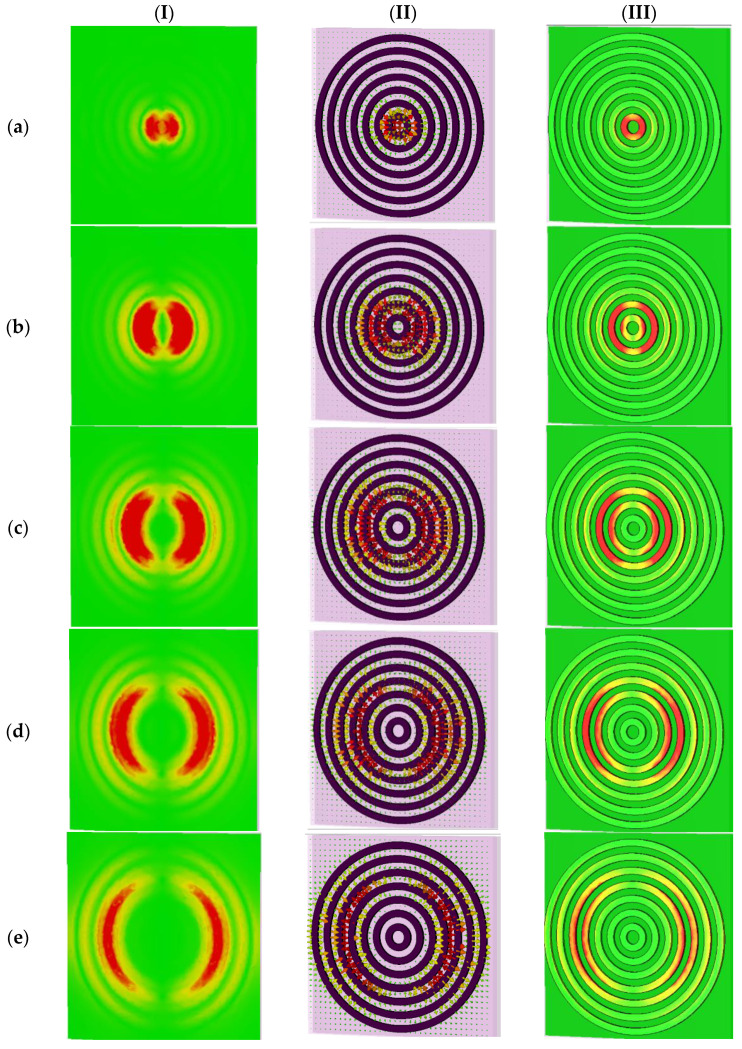
The electric field distributions and surface current energy densities on the top and bottom layer of the structure in XY-plane at different resonance frequencies. Column (I) shows electric field distributions at *Z* = 36 nm. Column (II) shows surface current electric energy densities. Column (III) shows electric field distributions at *Z* = 25 nm. For *unit cell G* at a frequency of: (**a**) $f_{1}=1.55$ THz, (**b**) $f_{2}=0.81$ THz, (**c**) $f_{3}=0.53$ THz, (**d**) $f_{4}=0.39$ THz, (**e**) $f_{5}=0.31$ THz, (**f**) $f_{6}=0.25$ THz, and (**g**) $f_{7}=0.2$ THz.

**Figure 7 sensors-22-02892-f007:**
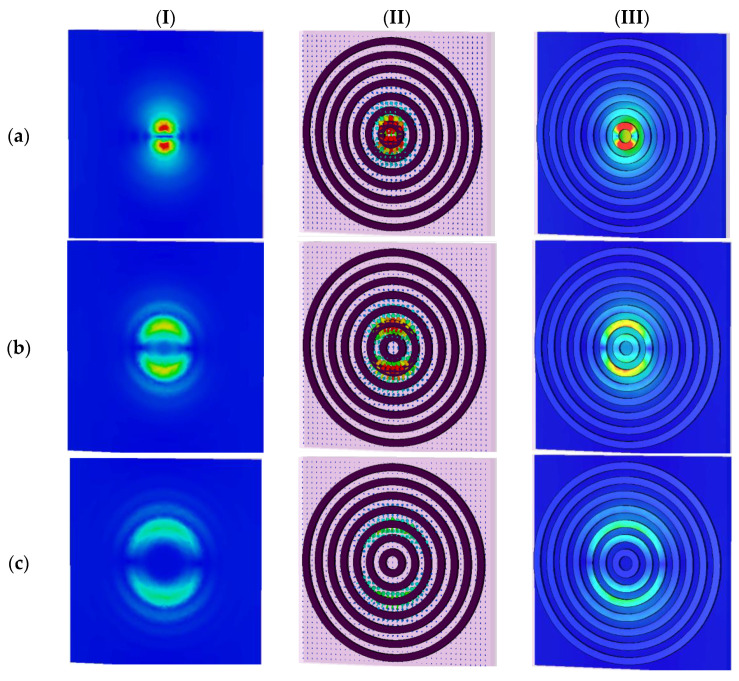
The magnetic field distributions and surface current densities on the top and bottom layer of the proposed microstructure in XY-plane at different resonance frequencies. Column (I) shows electric field distributions at *Z* = 36 nm. Column (II) shows surface current electric energy densities. Column (III) shows electric field distributions at *Z* = 25 nm. For *unit cell G* at frequency of: (**a**) $f_{1}=1.55$ THz, (**b**) $f_{2}=0.81$ THz, (**c**) $f_{3}=0.53$ THz, (**d**) $f_{4}=0.39$ THz, (**e**) $f_{5}=0.31$ THz, (**f**) $f_{6}=0.25$ THz, and (**g**) $f_{7}=0.2$ THz.

**Figure 8 sensors-22-02892-f008:**
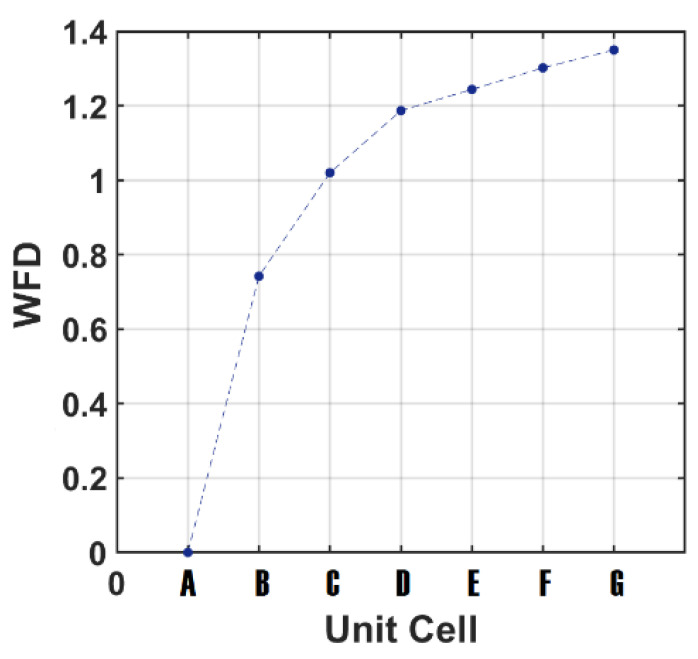
The working frequency duration (WFD) vs the number of RR antennas for different *unit cells* from *A* to *G*.

**Figure 9 sensors-22-02892-f009:**
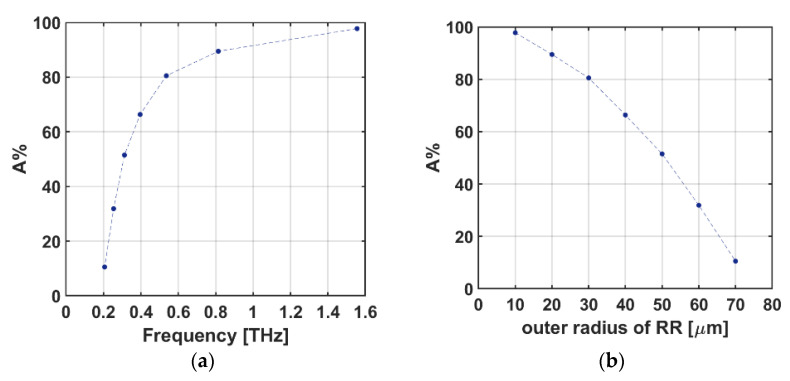
The absorption percentage of *unit cell G* vs (**a**) frequency, and (**b**) radius of the RR antennas.

**Table 1 sensors-22-02892-t001:** The geometrical parameters of the designed unit cells.

Parameters	$t_{b}$	$t_{s}$	$t_{m}$	$t_{RR}$	$r_{1}$	$r_{2}$	$r_{3}$	$r_{4}$	$r_{5}$	$r_{6}$	$r_{7}$
Value (µm)	8	5	20	2	10	20	30	40	50	60	70

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
