# Peer review of "Cost-Effective Bull’s Eye Aperture-Style Multi-Band Metamaterial Absorber at Sub-THz Band: Design, Numerical Analysis, and Physical Interpretation"

_sensors, 2022, doi:10.3390/s22082892_

Round 1

Reviewer 1 Report

In this work, the authors presented theoretical and numerical results for THz absorbing metasurface bull’s eye aperture style design with single- and N-beam absorption peaks. By aid of electromagnetic coupling with multiple semiconductor ring resonators, they can tune the number of the absorption peaks between 0.1-1.0 THz frequency. By adding the number of semiconductor rings without altering all other parameters, they found a translation trend of the absorption frequencies. Moreover, all spectral response results have also been explained by EM field distributions and surface currents. This topic is interesting and the work could be utilized in optical sensing and high-power THz sources. The main results are clearly presented. In my opinion, this paper deserves publication in Sensors, if the authors reasonably address the following comments:

  1. The level of English should be improved, as many sentences are badly constructed.
  2. There have been some relevant reports on the excitation of surface plasmon resonances and magnetic plasmon resonances in metamaterials and their related applications, especially for optical sensing, such as Journal of Lightwave Technology 39(13), 4525 (2021);  It should be properly cited in the introduction part to give audience a broader picture of this field and improve the quality of your paper.

Author Response

...

Reviewer 2 Report

This study discusses the spectral properties of a structure consisting of several concentric rings to sustain pronounced resonant states across the THz spectra. Extensive numerical computations have been utilized to extract and investigate the spectral properties of the designed subwavelength structure. It is demonstrated that the proposed platform can be tailored to support multiple resonant states across the THz band. The work drastically suffers from various drawbacks and cannot be accepted in the current format. The author should conduct a set of comprehensive revisions to improve the quality of the manuscript. I listed my comments in two categories below:

General comments:

1) What does "cost-effective" in the title of the work refer to? 

2) The writing style of the manuscript is poor and must be improved through conducting a set of comprehensive grammatical and writing style revisions. 

3) The spectra between 0.1 to 1 THz is well-known for the sub-THz band. The author must correct the associated sections.

4) There are up to 10 self-citations. The author must consider the state of the art studies that have been published in the THz concept, and focus on the strategic applications of the THz structures and metadevices. See: Materials Today 32, 108-130 (2020), Advanced Materials, 32(35), 2000250 (2020).

5) Bull's eye structures have extensively been analyzed in recent years. There should be a section in the bibliography to discuss the unique properties of this platform along with its applications. 

6) The Abstract is too general and does not emphasize the novelty of the work. This part must be revised carefully and there must be a stress on the impact of the conducted study. 

7) The employed fonts, label sizes, and formats are out of a scientific article.

8) The acronyms and abbreviations must be expressed one time. However, their complete format can be seen numerous times along the manuscript, and even in the Conclusions.

9) What does "devices structures" mean?

Specific comments:

10) The sharpness of the induced lineshapes in the Absorption spectra is interesting, however, the Q-factor, dephasing time, and modal volume of the induced resonant states have not been quantified. 

11) Cross-sectional magnetic-field profiles in vectorial style must be provided to understand how the electromagnetic waves concentrated between the upper resonator and mirror. 

12) What type of numerical package was employed to extract the spectral response of the structure? It looks the CST Microwave Studio program was utilized here, if so, the name if the software must be included/cited and the simulations details must be described. If not, the name of software should be provided. 

13) The first paragraph of Section 4 is a full fiasco and provides a sort of random details of surface plasmons without any focused purpose. Also, it is claimed that surface plasmons were generated within the structure and distributed along the rings. However, the influence of plasmons here is not clear to me. This must be argued. 

Author Response

Author’s Response to the Review Comments

Journal:                                      Sensors (ISSN 1424-8220)

Manuscript ID:        sensors-1662637

Title of Paper:          Cost-Effective Bull’s Eye Aperture Style Multi-Band Metamaterial Absorber at Sub-THz band: Design, Numerical Analysis, and Physical Interpretation

Author:                                     Zohreh Vafapour

Corresponding Author:  Zohreh Vafapour

Date Submitted:             March 17, 2022

Emails:                        [email protected]; [email protected] and [email protected]

Dear Editor,

Thanks for your letter and the reviewers’ comments in response to our manuscript to Sensors Journal. We really admire the processing of this Journal. We are really grateful to the anonymous reviewers for their valuable comments and suggestions on our manuscript. We have carefully read the reviewers’ report on our manuscript. Following all the reviewers’ comments and suggestions, the revised manuscript is improved and enriched greatly. We have benefited a lot from their professional suggestions and comments, believing that the revised manuscript can convey very well the results and novelty to the broad readership of this Journal. Many thanks for your hard work. The modified version with any changes noted by red font is named by “Revised Manuscript”. The itemized responses to the reviewer’s comments are listed in the following pages of this cover letter. We also attached the responses to reviewers’ comments as a separate file named “Response Letter”. We would be most grateful if our work could be accepted for publication in Sensors Journal.

Thank you very much for your time on our manuscript again! We appreciate the time and efforts of the editor and reviewers in reviewing this manuscript. We addressed all issues indicated in the review report and believed that the revised version can meet the journal publication requirements.

  • Important Note: The red sentences were added to the revised manuscript.

Response to Comments from Reviewer #1

In this work, the authors presented theoretical and numerical results for THz absorbing metasurface bull’s eye aperture style design with single- and N-beam absorption peaks. With aid of electromagnetic coupling with multiple semiconductor ring resonators, they can tune the number of the absorption peaks between 0.1-1.0 THz frequency. By adding the number of semiconductor rings without altering all other parameters, they found a translation trend of the absorption frequencies. Moreover, all spectral response results have also been explained by EM field distributions and surface currents. This topic is interesting, and the work could be utilized in optical sensing and high-power THz sources. The main results are clearly presented. In my opinion, this paper deserves publication in Sensors, if the authors reasonably address the following comments:

Comment #1:

1) The level of English should be improved, as many sentences are badly constructed.

Response:

We have greatly appreciated the professional opinion of the first reviewer made on our manuscript. Thank you for your attention to our paper and for considering it valuable research. Based on the reviewer's request, we checked all English parts of the paper and fixed every problem that is not an understandable sentence in the revised version with the red color of the manuscript.

Comment #2:

2) There have been some relevant reports on the excitation of surface plasmon resonances and magnetic plasmon resonances in metamaterials and their related applications, especially for optical sensing, such as the Journal of Lightwave Technology 39(13), 4525 (2021). It should be properly cited in the introduction part to give the audience a broader picture of this field and improve the quality of your paper.

Response:

We are greatly appreciated the time and effort the first reviewer made on our manuscript. Thank you for your attention to our paper. Based on the reviewer’s suggestion, we added some more relevant research work and cited some papers about the excitation of surface plasmon resonances and magnetic plasmon resonances in metamaterials and their related applications, especially for optical sensing such as the below references in the revised version of the manuscript as follows:

  1. Chen, J., Kuang, Y., Gu, P., Feng, S., Zhu, Y., Tang, C., Guo, Y., Liu, Z. and Gao, F., 2021. Strong magnetic plasmon resonance in a simple metasurface for high-quality sensing. Journal of Lightwave Technology, 39(13), pp.4525-4528.
  1. Ji, Y., Tang, C., Xie, N., Chen, J., Gu, P., Peng, C. and Liu, B., 2019. High-performance metamaterial sensors based on strong coupling between surface plasmon polaritons and magnetic plasmon resonances. Results in Physics, 14, p.102397.
  1. Chen, J., Peng, C., Qi, S., Zhang, Q., Tang, C., Shen, X., Da, H., Wang, L. and Park, G.S., 2018. Photonic microcavity-enhanced magnetic plasmon resonance of metamaterials for sensing applications. IEEE Photonics Technology Letters, 31(2), pp.113-116.
  1. Chen, J., Fan, W., Zhang, T., Tang, C., Chen, X., Wu, J., Li, D. and Yu, Y., 2017. Engineering the magnetic plasmon resonances of metamaterials for high-quality sensing. Optics express, 25(4), pp.3675-3681.
  1. Wang, B., Yu, P., Wang, W., Zhang, X., Kuo, H.C., Xu, H. and Wang, Z.M., 2021. HighQ Plasmonic Resonances: Fundamentals and Applications. Advanced Optical Materials, 9(7), p.2001520.
  1. Jeong, W.J., Bu, J., Jafari, R., Rehak, P., Kubiatowicz, L.J., Drelich, A.J., Owen, R.H., Nair, A., Rawding, P.A., Poellmann, M.J. and Hopkins, C.M., 2022. Hierarchically Multivalent Peptide–Nanoparticle Architectures: A Systematic Approach to Engineer Surface Adhesion. Advanced Science, 9(4), p.2103098.

We added all the new references in red color in the revised version of the manuscript.

Response to Comments from Reviewer #2

This study discusses the spectral properties of a structure consisting of several concentric rings to sustain pronounced resonant states across the THz spectra. Extensive numerical computations have been utilized to extract and investigate the spectral properties of the designed subwavelength structure. It is demonstrated that the proposed platform can be tailored to support multiple resonant states across the THz band. The work drastically suffers from various drawbacks and cannot be accepted in the current format. The author should conduct a set of comprehensive revisions to improve the quality of the manuscript. I listed my comments in two categories below:

General comments:

Comment #1:

1) What does "cost-effective" in the title of the work refer to?

Response:

We are greatly appreciated the professional comment and suggestion the second reviewer made on our work. Thank you for your attention to the article and for giving valuable suggestions. "cost-effective" refer to using a semiconductor material that is economic for the fabrication process with great results for mentioned application. As we explained in the manuscript, we used Indium antimonide (InSb) which is an economical/low-cost and commercially accessible material in the form of bulk padding and can be effortlessly found in extremely pure form for experimental fabrication of the proposed device.

Comment #2:

2) The writing style of the manuscript is poor and must be improved by conducting a set of comprehensive grammatical and writing style revisions.

Response:

We have greatly appreciated the professional opinion of the first reviewer made on our manuscript. Thank you for your attention to our paper and for considering it valuable research. Based on the reviewer's request, we checked all English parts of the paper and fixed every problem that is not an understandable sentence in the revised version with the red color of the manuscript.

Comment #3:

3) The spectra between 0.1 to 1 THz is well-known for the sub-THz band. The author must correct the associated sections.

Response:

Thank you for the valuable comment by the reviewer. We really appreciate it and changed all expressions “under/below 1 THz” to the expression “sub-THz band” in the revised version of the manuscript. Moreover, we change the title as well to your suggested expression follows:

Cost-Effective Bull’s Eye Aperture Style Multi-Band Metamaterial Absorber at Sub-THz band: Design, Numerical Analysis, and Physical Interpretation

Comment #4:

4) There are up to 10 self-citations. The author must consider the state-of-the-art studies that have been published on the THz concept and focus on the strategic applications of the THz structures and meta devices. See: Materials Today 32, 108-130 (2020), Advanced Materials, 32(35), 2000250 (2020).

Response:

We have greatly appreciated the reviewer for his/her attention to our research article. Really appreciate your time to review our work. Based on the reviewer’s suggestion, we added some more relevant research work and cited to some papers about the THz concept and their related applications such as the below references in the revised version of the manuscript as follows:

  1. Ahmadivand, A., Gerislioglu, B., Ahuja, R. and Mishra, Y.K., 2020. Terahertz plasmonics: The rise of toroidal metadevices towards immunobiosensings. Materials Today, 32, pp.108-130.
  1. Lee, S., Baek, S., Kim, T.T., Cho, H., Lee, S., Kang, J.H. and Min, B., 2020. Metamaterials for enhanced optical responses and their application to active control of terahertz waves. Advanced Materials, 32(35), p.2000250.
  1. Pitchappa, P., Kumar, A., Singh, R., Lee, C. and Wang, N., 2021. Terahertz MEMS metadevices. Journal of Micromechanics and Microengineering. 31 113001.
  1. Lou, J., Liang, J., Yu, Y., Ma, H., Yang, R., Fan, Y., Wang, G. and Cai, T., 2020. Siliconbased terahertz metadevices for electrical modulation of Fano resonance and transmission amplitude. Advanced Optical Materials, 8(19), p.2000449.
  1. Li, Y., Lv, J., Gu, Q., Hu, S., Li, Z., Jiang, X., Ying, Y. and Si, G., 2019. Metadevices with Potential Practical Applications. Molecules, 24(14), p.2651.

Comment #5:

5) Bull's eye structures have extensively been analyzed in recent years. There should be a section in the bibliography to discuss the unique properties of this platform along with its applications.

Response:

We have greatly appreciated the professional suggestions the reviewer made on our study. Thank you for your attention to the article. In the bibliography part of the submitted manuscript, there are detailed explanations about multi band THz structures and bull’s eye style in the manuscript. And based on the reviewers’ suggestion, we added some explanation and great references about using Surface plasmons in Bull's eye structures and using plasmonics and metamaterials physics to enhancements of the optical responses in the introduction part of therevised version of the manuscript as follows:

The potential of bull’s eyes as device elements has been well recognized. Some work has been performed to understand how such bull’s eye structures enhance and focus the optical properties of the system. In this study, we focused on excitation of electric and magnetic surface plasmons in the proposed bull’s eyes style device to achieve high absorption at different frequencies at sub-THz band which has potential applications in sensitive and selective optical biomedical sensing and detection, THz sources, etc.

Here are the new added references about BEA style structures in the revised version of the manuscript:

  1. Mahboub, O., Palacios, S.C., Genet, C., Garcia-Vidal, F.J., Rodrigo, S.G., Martin-Moreno, L. and Ebbesen, T.W., 2010. Optimization of bull’s eye structures for transmission enhancement. Optics express, 18(11), pp.11292-11299.
  1. Deng, X., Oda, S. and Kawano, Y., 2016, September. Split-joint bull's eye structure with aperture optimization for multi-frequency terahertz plasmonic antennas. In 2016 41st International Conference on Infrared, Millimeter, and Terahertz waves (IRMMW-THz) (pp. 1-2). IEEE.

Comment #6:

6) The Abstract is too general and does not emphasize the novelty of the work. This part must be revised carefully and there must be a stress on the impact of the conducted study.

Response:

We are greatly appreciated the reviewer for his/her attention to our research article. Really appreciate your time to review our work. Based on the reviewer’s suggestion, we changed the Abstract part and stressed on the important features of this proposed structure as well in the revised version of the manuscript as follows:

“Theoretical and numerical studies were conducted on plasmonic interactions at a polarization-independent semiconductor-dielectric-semiconductor (SDS) sandwiched layer design and a brief review of the basic theory model was presented. The potential of bull’s eye aperture (BEA) structures as device elements has been well recognized in multiband structures. In addition, the Sub-terahertz (THz) band (below 1 THz frequency regime) is utilized in Communications and Sensing applications which is in high demand in nowadays technology. Therefore, we produced theoretical and numerical studies for a THz absorbing metasurface BEA style design with N-beam absorption peaks at sub-THz band using an economical and commercially accessible materials which has low-cost and easy fabrication process. As well, we have applied the Drude model for the dielectric function of semiconductors due to its ability to describe both free electron and bound systems simultaneously. Associated with metasurface research and applications, it is essential to facilitate metasurface designs to be of utmost flexible properties with low-cost. By aid of Electro-magnetic (EM) coupling using multiple semiconductor ring resonators (RRs), we could tune the number of the absorption peaks between 0.1–1.0 THz frequency regime. By adding the number of semiconductor rings without altering all other parameters, we found a translation trend of the absorption frequencies. In addition, we validated our spectral responses results using EM field distributions and surface currents. Here, we mainly discussed the source of N-band THz absorber and the underlying physics of the multi-beam absorber designed structures. The proposed micro-structure has ultra-high potentials to utilize in high-power THz sources and optical biomedical sensing and detection applications based on opto-electronics technology based on having multiband absorption responses.”

Comment #7:

7) The employed fonts, label sizes, and formats are out of a scientific article.

Response:

We are greatly appreciated the professional suggestions the reviewer made on our study. We checked all the format, and fonts, etc and fixed any problems in the manuscript. We made all stuffs based on the journal sample in the revised version of the manuscript.

Comment #8:

8) The acronyms and abbreviations must be expressed one time. However, their complete format can be seen numerous times along the manuscript, and even in the Conclusions.

Response:

We are greatly appreciated the reviewer attention to our article. Really appreciate your time to review our work. We checked all acronyms and abbreviations in the revised version of the manuscript and fixed all problems in the revised version and showed them in red color.

Comment #9:

9) What does "devices structures" mean?

Response:

We are greatly appreciated the professional suggestions the reviewer made on our study. We fixed the expression to the correct one as “designed structures” in the revised manuscript.

Comment #10:

10) The sharpness of the induced lineshapes in the Absorption spectra is interesting, however, the Q-factor, dephasing time, and modal volume of the induced resonant states have not been quantified.

Response:

We are greatly appreciated the reviewer attention to our research article. Based on in detailed explanation of the concept of the work, we mainly focused, analyzed, and calculated the related parameters to explain the underlying physics of the phenomenon and creating multi (N-) band absorber in this research using Plasmonics, and electric and magnetic surface plasmons. And just introduced the potential applications of this research. We did not calculate Q-factor and other parameters which is not related to the topic of this research because the mainly focus of this study is on FUNDAMENTAL part of having THZ multiband design. Calculation of all these parameters should be a main topic of the new/future research which is not in this article. Thank you again for your time to review our work. ?

Comment #11:

11) Cross-sectional magnetic-field profiles in vectorial style must be provided to understand how the electromagnetic waves concentrated between the upper resonator and mirror.

Response:

We have greatly appreciated the professional suggestions the reviewer made on our study. Thank you for your attention to the article. Thank you for your suggestion as we discussed in Comment#10 about the main topic of this study, we mainly focused on electric and magnetic SURFACE plasmons and there is no effect for the Cross-sectional magnetic field between the upper resonator and mirror layer to be important here because they are not considered as surface. So, the future reader of the paper will be confused about Surface Plasmon understanding and their effect if the Cross-sectional magnetic field profiles between the upper resonator have had been added in the paper. Because these are not related to the underlying physics of this research. Again, really appreciate your time to read and review the paper. We are so grateful for your time to read and review our work. ?

Comment #12:

12) What type of numerical package was employed to extract the spectral response of the structure? It looks like the CST Microwave Studio program was utilized here, if so, the name of the software must be included/cited and the simulation details must be described. If not, the name of the software should be provided.

Response:

We have greatly appreciated the reviewer for his/her attention to our research article. Really appreciate your time to review our work. Based on the reviewer’s suggestion, we added the name of the simulator software and also added the references about the FDTD method for simulation details as well as scattering parameters in the revised manuscript and cite the related references.

  1. Introduction to the finite-difference time-domain (FDTD) method for electromagnetics. Synthesis Lect Comput Electromagn 2011, 6(1):1–250.

doi:10.2200/S00316ED1V01Y201012CEM027

  1. Resonant reflection by microsphere arrays with AR-quenched Mie scattering. Optics Express 2021, 29(12), pp.19183-19192.
  1. Disappearance of plasmonically induced reflectance by breaking symmetry in met-amaterials. Plasmonics 2017, 12(5), pp.1331-1342.

Comment #13:

13) The first paragraph of Section 4 is a full fiasco and provides a sort of random details of surface plasmons without any focused purpose. Also, it is claimed that surface plasmons were generated within the structure and distributed along with the rings. However, the influence of plasmons here is not clear to me. This must be argued.

Response:

We have greatly appreciated the professional suggestions the reviewer made on our study. Thank you for your attention to the article. We have already explained in detail the excitation of surface plasmon mode in the submitted manuscript as follows on page 12:

“As can be seen from Fig. 6, we plotted the distributions of the electric field, and the excited surface plasmons as surface current energy densities of the proposed microstructure in XY-plane at different resonance frequencies. Column I in Fig. 6 shows the electric field distribution at Z=36 nm.; Column II shows the excited surface plasmons as surface current electric energy densities; and columns III shows the electric field distributions at Z=25 nm. As clearly observable in row (a) of Fig.6, the innermost and smallest ring with a radius of is strongly coupled to the electric field at the mode of 1.55 THz. It supplies an independent electric dipole response (see Fig. 6a, columns I to III), and the surface charge oscillates is most along the sides driven by the external electric field. Therefore, it can be concluded that in this case, the mode and its correspondence absorption peak resonance are mostly due to the excitation of the RR antenna. The magnetic component of the incident wave penetrates between the top and bottom layers and generates antiparallel surface current on the RR antennas and the ground semiconductor plane, leading to the magnetic coupling and the response (see Figs. 7a columns I to III).

Figure 6 reveals that the absorption peak resonances originate from the dipole electric response of the RR antennas and the magnetic surface plasmon response between the air-Semiconductor-Dielectric (ASD) sandwiched layer. The first, second, third to finally seventh absorption peak resonance is associated with the EM resonance of the inner ring of, ,  to the outer ring of, respectively. The resonant frequencies are inversely proportional to the radius of RR antenna  (i = 1 to 7), which indicates that the bigger the radius of RR is the smaller the resonant frequency will be.”

But based on the reviewer’s suggestion, we added the name surface plasmon and defined it to be more understandable.

Once again, we would like to express our special thanks to the reviewers for their detailed and useful comments on the manuscript.

Corresponding author: Zohreh Vafapour

([email protected] & [email protected] &  [email protected])

Queen’s University, Electrical and Computer Engineering Department, Kingstone, Ontario, Canada

University of California, Merced, School of Natural Sciences, Department of Physics, Merced, California, USA

Round 2

Reviewer 2 Report

The suggested corrections and changes have been applied and addressed. Therefore, the work can be published as is.